# Microbial Inoculant GB03 Increased the Yield and Quality of Grape Fruit Under Salt-Alkali Stress by Changing Rhizosphere Microbial Communities

**DOI:** 10.3390/foods14050711

**Published:** 2025-02-20

**Authors:** Hao-Kai Yan, Cong-Cong Zhang, Guo-Jie Nai, Lei Ma, Ying Lai, Zhi-Hui Pu, Shao-Ying Ma, Sheng Li

**Affiliations:** 1College of Horticulture, Gansu Agricultural University, Lanzhou 730070, China; yanhaokai1995@163.com (H.-K.Y.); zc919634096@163.com (C.-C.Z.); naiguoj@st.gsau.edu.cn (G.-J.N.); 2Agronomy College, Gansu Agricultural University, Lanzhou 730070, China; malei2@gsau.edu.cn; 3College of Life Science and Technology, Gansu Agricultural University, Lanzhou 730070, China; ly182281@163.com (Y.L.); 13679424051@163.com (Z.-H.P.); 4Experimental and Base Management Center, Gansu Agricultural University, Lanzhou 730070, China; 5State Key Laboratory of Aridland Crop Science, Lanzhou 730070, China

**Keywords:** GB03 microbial agents, salt-alkali stress, soil rhizosphere microorganisms, berry quality, volatile substances in berries

## Abstract

Beneficial microbial agents, renowned for their cost-effectiveness, high efficiency, and environmental sustainability, play a pivotal role in enhancing plant growth, crop yield, and tolerance to abiotic stresses. This research delves into the impact of the GB03 microbial agent on the fruit quality of ‘*Cabernet Sauvignon*’ grapes, as well as on soil physicochemical properties and microbial communities under saline stress. The findings revealed that salt-alkali stress significantly elevated soil electrical conductivity, pH, Na^+^ levels, and total salt content, while it markedly reduced soil K^+^, organic matter, ammonium nitrogen, and nitrate nitrogen levels compared to the control. The application of the GB03 microbial agent, however, successfully mitigated these detrimental effects of salt-alkali stress. Furthermore, it augmented the population and abundance of dominant soil bacteria, including *Acidobacteriota*, *Bdellovibrionota*, and *Gemmatimonadota* etc., under saline conditions. Crucially, the microbial agent also inhibited the salt-alkali stress-induced decline in grape fruit’s single cluster weight, 100-grain weight, fruit color intensity, and volatile aroma compounds, as well as the increase in organic acids. Consequently, the GB03 microbial agent emerges as a potent strategy for ameliorating saline-alkali soils and bolstering the salt-alkali stress resilience of horticultural crops like grapes.

## 1. Introduction

The grape (*Vitis vinifera* L) is one of the most widely cultivated and economically important berry crops in the world. About 71% of the grape yield is used for wine, 27% for fresh berry and 2% for dried berry. Therefore, wine grapes have received widespread attention and cultivation. ‘*Cabernet Sauvignon*’ grapevines originally from the Bordeaux region of France and widely cultivated worldwide, was first imported and cultivated in China in 1892 [1,2], which has now become the largest cultivated red wine variety in China. The berries of ‘*Cabernet Sauvignon*’ grapes are small, acidic, dark blue, thick-skinned, and aromatic, which make it the first choice for many famous wineries to produce high-quality red wines [3].

For wine grapes, berry quality is influenced by a variety of factors, including variety, soil, climatic conditions, maturity and cultivation system [4,5]. In particular, soil pH, nutrient availability and microbiological influences are critical to berry development and quality [6,7]. Research has shown that an unsuitable soil pH limits the uptake of nutrients by grape roots, which leads to a deterioration in the quality of the berries [8]. Nitrogen and potassium in the soil are also closely related to the growth and quality of the grapes. Changes in soil nitrogen affect anthocyanins, flavonoids [9], total soluble solids (TSS) and titratable acids [10,11]. Potassium contributes to the sugar accumulation of the berry and nutrient transport [12]. Soil microbial communities (bacteria, fungi, nematodes, etc.) are microscopic in the rhizosphere microenvironment, directly or indirectly affecting crop health and yield [13]. Soil microbial sulfur metabolism may play a key role in shaping plant physiology, grape and wine quality suggesting soil microorganisms are involved in the growth and development of grape plants and berries [7]. However, with regard to soil microorganisms, information on the effect of soil microbiota on berries is still lacking.

In recent years, with global warming and reduced rainfall, grape cultivation areas located in arid and semi-arid regions have faced harsh soil salinization problems [14]. Soil salinization not only directly leads to a large loss of soil carbon pools, but also is an important cause of alkali-hydrolyzable nitrogen deficiency [15,16]. The high salinity and pH of saline soil affect the conversion of soil carbon and nitrogen, which leads to the reduction of organic matter and nutrient content, and the decrease of soil fertility [17,18,19]. Research has found that soil related enzyme activity, such us soil chitinase, sucrase and oxidase, gradually decreased with the increasing of salt concentration [20,21,22]. Salt can indirectly affect soil nitrogen mineralization and transformation by influencing microbial activity, which in turn leads to a decrease in organic matter, nutrient content, and soil fertility [23]. Therefore, the high salinity and high pH of saline soils affect the conversion of soil carbon and nitrogen. Grapes belong to salt tolerant and neutral sensitive berry trees, and salinity is a severe problem for vegetative growth in grapes. Salinity reduced the yield components (each bud bunch, weight per bunch, weight per berry, and total yield) of the self-rooted Sudanese grapes [24]. Therefore, exploring effective methods to alleviate the adverse effects of saline alkali soil on crops such as grapes has always been a focus of attention.

The application of Plant Growth Promoting Rhizobacteria (PGPR) has been recognized as an effective measure to increase agricultural productivity and improve soil and environmental health [25]. PGPR may improve soil and increase crop tolerance to salt-alkali stress, thus promoting plant growth, and improving crop yield and quality [26,27]. The application of *Kosakonia radicincitans* (DSM 16656) could alter the amino acids, sugars, and volatile components of ripe tomato fruit, making the fruit taste more delicious [28]. After applying the microbial agent to wheat (Jimai22), both wheat yield and available nitrogen content in the soil significantly increased [29]. Sarabia et al. found that rhizosphere yeast improved the growth of corn (DK-2061) shoots and roots, maintained plant health, and increased soil fertility [30]. Under saline conditions, the germination rate and growth of lettuce were promoted by applying a biofertilizer with pear-shaped spore algae extracts and PGPR (*Azospirillum brasilense*) [31]. *Bacillus subtilis* strain GB03 was isolated in 1971 from the lysed mycelium of Sclerotium rolfsii, which was found in wheat-field soil in Glen Osmond, South Australia. Subsequently, as a result of comprehensive taxonomic reclassification within the *Bacillus* genus, *B. subtilis* strain GB03 was redesignated as *B. amyloliquefaciens* strain GB03, which has now been formally renamed as *B. velezensis* strain GB03 [32,33]. *B. velezensis* strain GB03 continues to serve as a model bacterium for studying plant-bacteria interactions. PGPR *Bacillus velezensis* strain GB03, a well-studied and representative *Bacillus* strain, has the ability to protect plants against foliar pathogens, soil-borne pathogens, and abiotic stress. Additionally, GB03 could promote plant growth and increase crop yield [34]. Until now, studies have focused on the positive roles of microbial inoculant GB03 in stress tolerance in forage and Chinese herbal medicinal crops such as white clover [35] and *Codonopsis pilosula* [36]. Recently, Wang et al. [37] have shown that *Bacillus subtilis* GB03 promoted the growth of tall fescue by regulating plant hormones and nutritional homeostasis. ‘*Cabernet Sauvignon*’ grapes, as an important economic crop, have received widespread attention and cultivation worldwide. However, soil salinization poses a serious challenge to the wine grape industry. It is currently unclear whether microbial inoculant GB03 can change the physicochemical properties of grape soil, enzyme activity, microbial community diversity under salt-alkali stress, and improve grape fruit quality and aroma components. Therefore, we hypothesised that microbial inoculant GB03 could mitigate the adverse effects of salt alkali stress on soil physico-chemical properties and inter-root microorganisms, and that it could improve fruit yield and quality under stress conditions. The aim of this study was to evaluate the effect of microbial inoculant GB03 on the salt alkali tolerance of *Cabernet Sauvignon* grapes and to demonstrate the potential use of beneficial bacterial strains in grapevines under saline conditions.

## 2. Materials and Methods

### 2.1. Field Conditions and Materials

The experiment was conducted at the Mogao Wine Estate in the Hexi Corridor, China (102° 52′ E, 37° 50′ N) at an altitude of 1632 m. The soil is neutral to weakly alkaline gravelly sandy loam with deep soil layer and good air permeability, with an annual rainfall of 191.9 mm and an evaporation of 2130.8 mm, and the annual average sunshine hours are 2791.8 h, the effective accumulated temperature of ≥10 °C is 2800–3200 °C, and the annual average temperature is 7.6 °C. The climate of the region is warming temperate, with a frost-free period of 160 days. The underground soil is buried for winter, and there is sufficient sunlight during the production period. There is a large temperature difference between day and night [38]. The experimental field in this area is a north-south T-shaped frame, with 5-year-old self-rooted Cabernet Sauvignon wine grapes selected.

GB03 microbial agent is developed by Lanzhou University and Gansu Agricultural University, and produced by Gansu HongYuan Biotechnology Co., Ltd. (Lanzhou, China) (The main components are *Bacillus velezensis*, effective viable count ≥ 10^8^ CFU/g).

### 2.2. Experimental Procedure and Treatments

The ‘*Cabernet Sauvignon*’ vines are self-rooting (5 years old) with a row spacing of 2 m and a plant spacing of 0.5 m. Three rows of grapes with similar growth were selected, and the middle nine grapes were selected as one treatment for each row. Three field fertilization treatment groups were set up: CK (3 L 1:50 dilution of GB03 microbial agent + 3 L deionized water applied to each grape tree after inactivation), YJ (3 L 1:50 diluted GB03 microbial agent applied to each grape tree + 3 L 300 mmol/L NaCl + NaHCO_3_ mixed salt solution (The ratio of NaCl and NaHCO_3_ was 1:1. 8.766 g of NaCl and 12.601 g of NaHCO_3_ were fixed to 1 L with deionised water), pH > 8.5), Y (3 L 1:50 diluted GB03 microbial agent applied to each grape tree after inactivation + 3 L 300 mmol/L NaCl + NaHCO_3_ mixed salt solution (The ratio of NaCl and NaHCO_3_ was 1:1. 8.766 g of NaCl and 12.601 g of NaHCO_3_ were fixed to 1 L with deionised water), pH > 8.5). The grape tree roots in each group mainly distribute in a soil layer of 20–40 cm, with a fertilization depth of 30 cm. Circular fertilization was used around the grape tree roots in each treatment. The fertilization was processed a total of 4 times, including 15 May 2023 (flowering period), 15 June 2023 (berrying period), 15 July 2023 (swelling period), and 15 August 2023 (color transition period). In each treatment, salt solution treatment was carried out, and 10 d later, GB03 microbial agent was added. The pruning method and degree for grape trees, as well as measures for pest and disease control, were consistent with previous years in vineyard management.

### 2.3. Sample Collection

The grape fruit were harvested when they were ripe (25 September 2023), and the actual yield of each treatment was weighed, which is converted into a hectare yield. Nine clusters of berry clusters were taken from the shaded and sunny sides of each grape tree, and 1500 berries were randomly selected from the upper, middle, and lower parts of the clusters. The obtained berries were packaged on the same day and quickly transported back to the laboratory for storage at 4 °C for the determination of physiolgical and biochemical indicators and volatile aroma content.

Soil samples should be collected at a depth of 20~40 cm around the roots of the grapevine. The specific method for doing this is described in detail in the work of Zhang et al. [39]. The rhizosphere soil of the root system was brushed into a sterile centrifuge tube and immediately stored in liquid nitrogen. Three samples in each treatment were collected, each of which was a mixture of 5 sampling points [40].

### 2.4. Determination of Parameters

#### 2.4.1. Soil Physical and Chemical Indicators

Soil conductivity was determined using a conductivity meter (DDS-307, INESA scientific instrument Co., Ltd., Shanghai, China), pH was determined using a pH meter (PH600L, INESA scientific instrument Co., Ltd., Shanghai, China), and the content of soil Na^+^ and K^+^ was determined according to Devitt et al. [41]. The determination of total salt content followed the method of Pasković et al. [42], and organic matter content was determined according to Alhassan et al. [43]. The content of ammonium nitrogen and nitrate nitrogen was determined using the method of Hosseini et al. [44].

#### 2.4.2. Soil Enzyme Activity

The soil sucrase (SSC-1-Y), soil urease (SUE-1-Y), nitratase (NR-1-W), and nitrite reductase (SNIR-1-G) kits (Suzhou Keming Biotechnology Co., Ltd., Suzhou, China) were used to measure activities.

#### 2.4.3. Soil Microbial Parameters

The extraction of nucleic acids was done using the soil DNA kit (DP-812, Tiangen Biotechnology Co., Ltd., Beijing, China). Nucleic acid concentration was detected using an enzyme-linked immunosorbent assay (ELISA) reader. The extracted DNA was subjected to 16S full-length amplification for bacteria and ITS full-length amplification for fungi. The bacterial amplification primers were ′5′-AGRGTTTGATYNTGGCTCAG-3′, ′5′-TASGGHTTACCTGTTASGAGATT-3′, and the fungal amplification primers were 5′-CTTGGTCATTTAGGAGATAA-3′ and 5-GCTGCGTTCTTCATCGATGC-3″, respectively. After amplification, the product was purified and quantified with real-time polymerase chain reaction (PCR), and the text is completed by library builds. Library concentration and size were measured by Qubit and Agilent 2100, respectively, and then we advanced machine-based sequencing on a Sequel II sequencer.

The circular consumption sequencing (CCS) file was exported from the raw data of the offline machine, barcode recognition and length filtering were performed, chimeras on the CCS sequence were removed, and the effective CCS was obtained. The effective CCS sequence was clustered and denoised, OTUs was divided, and their species classification was obtained based on their sequence composition. Microbial diversity, species composition, and correlation analysis with environmental factors were performed.

Based on the results of OTU cluster analysis, the Alpha-diversity (QIME2 2020.6H) of the three samples, the community richness (Chao http://www.mothur.org/wiki/Chao, accessed on 16 April 2024. Ace http://www.mothur.org/wiki/Ace, accessed on 16 April 2024), the community diversity (Shannon http://www.mothur.org/wiki/Shannon, accessed on 16 April 2024. Simpson http://www.mothur.org/wiki/Simpson, accessed on 16 April 2024), and sequencing coverage (http://www.mothur.org/wiki/Coverage, accessed on 16 April 2024) were calculated [45].

#### 2.4.4. Yield of Grape Berry

Grapes from each vine were weighed to determine the fruit yield for each treatment. For single cluster weight, in each treatment, 10 clusters were randomly selected and weighed. For hundred-grain weight, 100 wine grapes were randomly selected during the grape ripening period to measure the hundred-grain weight of grape berry. An electronic balance (AE224 Electronic Analytical Balance, Shanghai Sunny Hengping Technology Instrument Co., Ltd., Shanghai, China) with an accuracy of 1/10,000 was used to determine single cluster weight and hundred-grain weight.

#### 2.4.5. Reducing Sugar, Soluble Solid, Fructose, Glucose, Sugar-Acid Ratio, Total Tannin, Titratable Acidity Content, and pH Value in Grape Berry

According to Acevedo-Opazo et al. [46], a multi-function Energy Wine Analyzer [Foss DK-3400, Foss (Beijing) Science and Trade Co., Ltd., Beijing, China] was used to determine the content of reducing sugars, soluble solids, fructose, glucose, and titratable acidity, and pH value. Sugar acid ratio (SAR) is the ratio of soluble solid to titratable acidity [47].

After each sample was weighted, the skins and seeds were manually separated, washed several times with deionized water and subsequently dried by absorbent papers, then extracted in 20 mL of 2:1 acetone/water for 24 h in the dark. The solutions were filtered to exclude solid tissues via a Büchner funnel. The skin extraction was used for total tannins. Total tannins were analyzed according to the study from Zhang et al. [48].

#### 2.4.6. Grape Berry Color

The L*, a*, and b* values of the berry epidermis were measured using the high-precision color-reader (CR-10, Konica Minolta, Inc., Tokyo, Japan) and the C* values were calculated with 3 replications. Here, L* represents brightness; a* represents red-green values, in which positive values represent red and negative value represents green; b* represents yellow blue value, in which positive value represents yellow and negative value represents blue; C* represents color saturation (the larger the value, the purer the color) [49].

#### 2.4.7. Anthocyanins Content in Grape Skins

The buffer solution was prepared with 5 g of tartaric acid/L, 12% (*v*/*v*) ethanol, 2 g of sodium metabisulphite/L, and then was buffered at pH 3.2 using 1 mol/L NaOH. For each treatment, three replicates with 15 berries each were weighted and used for the extraction. Skins were manually separated from the pulp, then quickly inserted in 40 mL of the extracting solution, and subsequently frozen. Samples were homogenized with an immersion blender (EF500-T, OuHor Machinery Equipment Co., Ltd., Shanghai, China), and centrifuged at 4200× *g* at 20 °C for 5 min. The supernatant was diluted to 50 mL of total volume using the same buffer solution, and was used for anthocyanin analysis [50].

#### 2.4.8. Total Phenol Content in Grape Berry Skins

The content of total phenols was determined by the Folin Schottgart method. Grape skin sample (1 g) was weighed and methanol hydrochloride was added to avoid light. The supernatant was taken for testing after centrifugation, and the ultraviolet fraction was used after adding forint-shoka and sodium carbonate water bath for 2 h. The UV Visible Spectrophotometer (TU-1810, Beijing Puxi General Instrument Co., Ltd.; Beijing, China) was used to record the absorbance colorimetrically at 765 nm, and the total phenolic content was expressed as the dry mass of gallic acid per gram of sample [51].

#### 2.4.9. Volatile Substance Content in Grape Berry

Extraction and determination of volatile compounds: Sample (5 g) was weighed, and 0.05 g of polyvinylpyrrolidone was added. 0.025 g D-gluconic acid, 2 g NaCl, 10 µL internal standard (2-octanol, 81.06 mg/L) and a magnetic stirring rotor were mixed in a 20 mL sample vial. Constant temperature water bath (40 °C) with magnetic stirring was kept for 30 min, and the activated extraction tip was inserted into the headspace of the vial, stirred and heated at 40 °C for 30 min, and finally the tip was inserted into the gas chromatography inlet and was resolved at 240 °C for 8 min.

GC-MS conditions: Non split mode injection, injection port temperature of 240 °C; Heating program: Starting at 40 °C, keeping it hold for 5 min, then increasing to 220 °C at 3 °C/min and holding at 220 °C for 15 min. Then, we record the mass spectrometry retrieval of volatile substances in the range of 35–350 *m*/*z* under 70 eV electron collision mode.

Standard curve production: Using blank matrix (grape juice simulation solution: 7 g/L tartaric acid, 200 g/L glucose) as the solvent, the standard of each volatile substance component was formulated into a single standard of a certain concentration, and each single standard was prepared into a mixed standard of 5 concentration gradients. Blank matrix (5 g), 10 μL of standard mix, and 2 g of NaCl, and magnetic rotor were added to the headspace vial. Subsequently, extraction and detection were carried out under the same conditions, with 3 replications in each concentration gradient. A standard curve was established with standard concentration as the *x*-axis and standard peak area as the *y*-axis.

Qualitative and quantitative methods: The mass spectra detected in the sample were compared and qualitatively analyzed using NIST and Wiley databases combined with the peak time of standard samples. Aromatic compounds with standard samples were quantified according to the corresponding standard curve. For compounds without standard samples, the concentration of the compounds was calculated using semi quantitative method using 2-octanol (81.06 mg/L) [51].

### 2.5. Statistical Analysis

Excel 2021 was used for data statistics and SPSS Statistics 24.0 was used for one-way analysis of variance (ANOVA) to evaluate significant differences, and least significant difference (LSD) analysis was used to evaluate statistical differences between treatment (*p* < 0.05). Drawing was performed with Origin 2022 64 Bit software.

## 3. Results

### 3.1. Effects of GB03 Microbial Inoculant Treatment on Soil Physicochemical Properties and Soil Enzyme Activities in Vineyard Under Salt Alkali Stress

As shown in Figure 1, the electrical conductivity, soil pH, and Na^+^ and total salt content of the vineyard soil in Y treatment were higher than those in CK (*p* < 0.05). The electrical conductivity, soil pH, and Na^+^ and total salt content of the vineyard soil under YJ treatment were lower than those under Y treatment, with decreases of 2.01%, 3.85%, 24.72%, and 11.70% respectively. The electrical conductivity and total salt content were significantly lower in YJ treatment than in Y treatment (*p* < 0.05). Under Y Treatment, K^+^ content in vineyard soil, the content of organic matter, ammonium nitrogen, and nitrate nitrogen significantly decreased by 62.87%, 40.09%, 52.12%, and 57.72% compared with CK, respectively (*p* < 0.05). However, the content of K^+^, organic matter, ammonium nitrogen and nitrate nitrogen in the vineyard soil under YJ treatment was significantly higher than those under Y treatment, which were increased by 100.47%, 43.99%, 72.34% and 58.80%, respectively.

The soil sucrase activity of the vineyard under YJ treatment was significantly higher than that under CK and Y treatment (*p* < 0.05), with an increase of 4.12% and 4.44% respectively. In YJ treatment, the soil urease activity in vineyards was significantly higher than that in CK and Y treatment (*p* < 0.05), with an increase of 50.07% and 82.07%, respectively. The nitrate reductase activity in the soil under CK treatment was significantly higher than that under YJ and Y treatment (*p* < 0.05), which was increased by 166.67% and 405.88%, respectively. Compared to Y treatment, YJ treatment increased the nitrate reductase activity by 89.71%. There was no significant difference in soil nitrite reductase activity under the three treatments.

### 3.2. Effects of GB03 Microbial Agent Treatment on Soil Microbial Community α and β Diversity Under Salt Alkali Stress

Alpha diversity is a comprehensive indicator reflecting the richness and diversity of soil microorganisms. At a classification level of 97%, the diversity index of soil bacterial and fungal communities under different fertilization treatments was analyzed (Figure 2). Under Y treatment, the soil Ace, Chao, Simpson, and Shannon indices of soil bacterial microorganisms were significantly higher than those of CK (*p* < 0.05). Compared with Y treatment, the diversity index of YJ treatment showed a decreasing trend, but there was no significant difference between them. Compared with CK, Y treatment significantly increased the Simpson and Shannon indices of soil fungi, but had no significant effect on ACE and Chao. The diversity index of soil fungi after YJ treatment was significantly higher than that of CK and Y treatment (Figure 2; *p* < 0.05).

Based on the Binary jaccard distance between samples, principal component analysis was performed for soil bacteria and fungi under different treatments (Appendix A). It was found that the explanatory power of PC1 on the structure of bacterial and fungal communities was 80.85% and 73.95%, respectively; while the explanatory power of PC2 on the structure of bacterial and fungal communities was 14.50% and 24.15%, respectively (Appendix A).

The coverage in bacterial and fungal communities reached 95.35% and 98.10%, respectively (Appendix A), which could comprehensively reflect the sample information. Moreover, the dispersion distance between the three treatments was relatively long, indicating significant differences in the composition of bacterial and fungal microbial communities among the three treatments.

### 3.3. Differences in the Horizontal Distribution of Microbial Bacteria and Fungi Phyal in Vineyard Soil Under Salt Alkali Stress After GB03 Microbial Agent Treatment

We identified a total of 31 bacterial phyla and 11 fungal phyla (Appendix A). Compared with CK, the number of dominant bacteria and fungi increased (>1%) after Y treatment, while the number of dominant bacteria and fungi in YJ treatment was lower than that in Y treatment. For bacterial communities, *Acidobacteriota*, *Actinobacteriota*, *Bacteroidota*, *Bdellovibrionota*, *Gemmatimonadota*, *Planctomycetota*, *Proteobacteria* and *Verrucomicrobiota* were common dominant bacteria under different treatments. Among them, the relative abundance of *Acidobacteriota*, *Gemmatimonadota*, and *Bdelovibrionota* under Y treatment was significantly higher than that under CK, while the relative abundance of *Actinobacteriota* and *Bacteroidota* was significantly lower than that under CK. After YJ treatment, the relative abundance of *Acidobacterota* and *Gemmatimonadota* was significantly lower than that of Y treatment, while there was no significant difference in the rest. For fungal communities, *Ascomycota*, *Basidiomycota*, and *Mortierellomycota* were common dominant fungi under different treatments. Compared with CK, the relative abundance of *Ascomycota* and *Mortierellomycota* significantly increased under Y treatment, while *Basidiomycota* significantly decreased. There was no significant difference in the relative abundance of *Basidiomycota* between YJ and Y treatments, while the relative abundance of *Ascomycota* and *Mucoromycota* was significantly lower in Y treatment than in YJ treatment.

### 3.4. Differences in the Horizontal Distribution of Microbial Bacteria and Fungi Genera in Vineyard Soil Under Salt Alkali Stress After GB03 Microbial Agent Treatment

An inter group analysis of variance was conducted for the abundance of bacteria and fungi at the genus level in different treatments, and the top 10 species with the lowest *p*-value were displayed (Figure 3). As for bacteria, *Bacillus* had the highest relative abundance after YJ treatment, with significant differences compared to CK and Y treatment (Figure 3a). In addition, compared with CK, Y treatment increased the relative abundance of *Candidatus_Alysiosphaera*, *Pelagibius*, *Wandonia*, *unclassified_Alphaproteobacteria*, and *Acidibacter* in the soil. However, after YJ treatment, the relative abundance of these bacteria significantly decreased, and all bacteria, except for *unclassified_Alphaproteobacteria*, returned to the levels observed in the CK treatment. Compared with CK, Y treatment recruited bacterial communities such as *Aeromonas*, *Candidatus-Alysiosphaera*, and *Wandonia* in the rhizosphere soil of grapes, while significantly increasing the relative abundance of *Bacillus*, *Pelagibius*, *unclassified_Alphaproteobacteria*, *unclassified_Fibrobacteraceae*, and *Acidibacteria*, and significantly reducing the relative abundance of *Novosphingobium*. Under YJ treatment, the relative abundance of *Aeromonas*, *Bacillus*, and *UTBCD1* microbial communities significantly increased, while the relative abundance of *Pelagibius*, unclassified *Alphaprotobacteria*, and *Acidibacter* microbial communities significantly decreased, compared with Y.

In the fungal community (Figure 3b), Y treatment significantly increased the relative abundance of *Mortierella* and *Linatneria* compared with CK; while after YJ treatment, their relative abundance was significantly lower than that in Y treatment. These results indicate that microbial agents may help regulate soil microbial community structure under salt stress conditions by promoting the growth of certain beneficial microorganisms and inhibiting the increase of certain microbial populations under salt stress, thereby improving the soil environment. Compared with CK, Y treatment had a recruitment effect on fungal communities such as *Linatneria* and *Diehliomyces*, and the relative abundance of *Filobassidium* and *Mortierella* genera significantly increased, while the relative abundance of *Leucoagaricus* genera significantly decreased. Among the three treatments, *Acauropage* and *Alfaria* genera only appeared in the rhizosphere soil under YJ treatment. Compared with Y treatment, YJ significantly decreased the relative abundance of *Leucoagaricus* and *Mortierella* genera, while significantly increased the relative abundance of *Filobassidium*, *Graphium*, *Fusurium*, *Exophiala*, and *Diehliomyces* genera.

### 3.5. Correlation Analysis of GB03 Microbial Agent Treatment on Soil Microorganisms and Physicochemical Indexes in ‘Cabernet Sauvignon’ Vineyards Under Salt Alkali Stress

The correlation analysis results between soil bacterial and fungal content and soil physicochemical indexes are shown in Figure 4. There was a significant negative correlation between soil *Novosphingobium* and *Leucoagaricus* relative abundance and soil conductivity. The relative abundance of *unclassified_Alphaproteobacteria* was significantly positively correlated with soil pH and total salt content. The relative abundance of *Acauropage*, *Alfaria*, *Graphium*, and *Fusarium* was significantly positively correlated with the content of invertase; while there was a highly significant positive correlation between *Exophylla* relative abundance and invertase content. There was no significant correlation between the content of soil Na^+^, K^+^, organic matter, ammonium nitrogen, nitrate nitrogen, urease, nitratase, and nitrite reductase with the content of soil fungi and bacteria.

### 3.6. Effect of GB03 Microbial Inoculant Treatment on Phenotypic Characters and Yield of ‘Cabernet Sauvignon’ Grape Fruit Under Salt Alkali Stress

GB03 microbial inoculant treatment significantly alleviated the negative effects of salt alkali stress on grape fruits (Figure 5a). The single cluster weight (Figure 5b), hundred grain weight (Figure 5c), and yield of grape plants (Figure 5d) under Y treatment significantly decreased by 29.67%, 18.37%, 36.63%, respectively, compared with CK. Compared with Y treatment, the single spike weight, hundred grain weight, and yield of grape plants under YJ treatment were significantly increased by 27.01%, 11.63%, and 35.91%, respectively.
Figure 4Correlation analysis between soil microorganisms and physicochemical parameters in vineyards and grape quality and volatile matter indicators. 1–32 respectively represent: 1, *Aeromonas*, 2, *Bacillus*, 3, *Candidatus_Alysiosphaera*, 4, *Novosphingobium*, 5, *Pelagibius*, 6, *UTBCD1*, 7, *Wandonia*, 8, *unclassified_Alphaproteobacteria*, 9, *unclassified_Fibrobacteraceae*, 10, *Acidibacter*, 11, *Acaulopage*, 12, *Alfaria*, 13, *Filobasidium*, 14, *Graphium*, 15, Leucoagaricus, 16, *Mortierella*, 17, *Fusarium*, 18, *Exophiala*, 19, *Lindtneria*, 20, *Diehliomyces*, 21, Electrical conductivity, 22, Soil pH, 23, Soil Na^+^, 24, Soil K^+^, 25, Total salt, 26, Organic matter, 27, Ammonium nitrogen, 28, Nitrate nitrogen, 29, Invertase, 30, Urease, 31, Nitratase, 32, Nitrite reductase. Positive correlations are shown in red and negative correlations in blue, with smaller ellipses indicating higher correlations. The numbers in the graphs indicate the correlation coefficients, with positive values indicating positive correlations and negative values indicating negative correlations. * indicates significant correlation at the 0.05 level, ** indicates significant correlation at the 0.01 level.
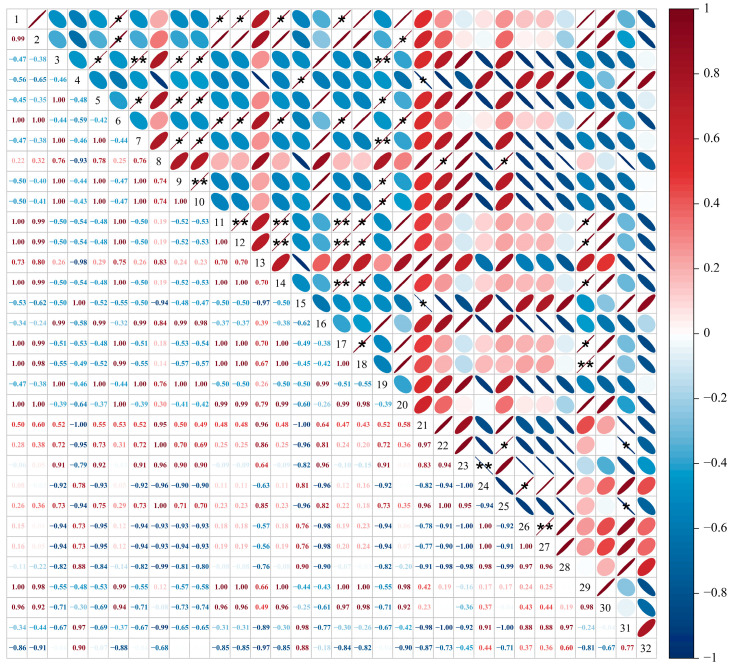



### 3.7. Effect of GB03 Microbial Inoculant Treatment on Fruit Quality of ‘Cabernet Sauvignon’ Grape Fruit Under Salt Alkali Stress

As shown in Figure 6, the content of TSS, fructose, glucose, reducing sugar, sugar-acid ratio, the content of total anthocyanin, and total tannin significantly decreased by 23.68%, 11.40%, 12.07%, 7.72%, 50.43%, 33.26% and 39.51%, respectively. Compared with Y treatment, YJ treatment significantly increased the content of TSS, fructose, glucose, reducing sugar, sugar-acid ratio, total anthocyanin and total tannin content by 15.85%, 9.68%, 9.63%, 7.34%, 33.04%, 31.56%, and 15.17%, respectively. The content of tartaric acid, malic acid, gluconic acid, lactic acid and titratable acid in Y treatment was significantly higher than that in CK (*p* < 0.05). However, the content of tartaric acid, malic acid, gluconic acid, lactic acid, and titratible acid in YJ treatment was lower than that in Y treatment. The lactic acid content in YJ treatment was significantly lower than that in Y treatment (*p* < 0.05). There was no significant difference in pH and Total phenol among the three treatments.

According to Table 1, there was no significant difference in the a*, b*, and C* values of grape berry among the three treatments, but the L* value in Y treatment was higher than that in YJ treatment and CK.
Figure 5Changes in phenotype and yield difference of grape fruit after GB03 microbial inoculant treatment. (**a**) Phenotypic characters, Scale bar: 2.5 cm × 2.5 cm; (**b**) Cluster weight, (**c**) 100-grain weight, (**d**) Yield. Values (means ± standard deviations) were calculated from three independent replicates. The bars in the graph are error lines. Lowercase letters represent significant differences among sampling treatments (*p* < 0.05).
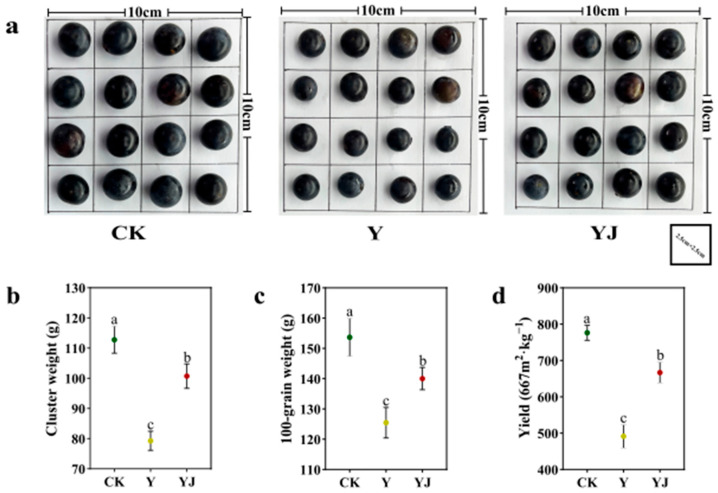



### 3.8. Partial Least Squares and Principal Component Analysis of Volatile Aroma Compounds in ’Cabernet Sauvignon’ Grape Fruit Under Salt Alkali Stress After Treatment with GB03 Microbial Inoculant

PLS-DA analysis on all volatile aroma compounds of grape berries (Appendix A) under different treatments was performed to obtain VIP score graphs. As shown in Figure 7, the importance of the volatile aroma compounds after each treatment was shown by color gradient where red indicates high levels of content and blue indicates low levels of content. The volatile aroma compounds in grape berry which were most affected were selected for each treatment, including hexan-1-ol, hexanal, 2-hexenal, trans-2-hexenal, 2,6-Di-tert-butyl-4-methylphenol, and Decyl Decanoate. Among them, C6 aroma compounds were found to be more abundant. Meanwhile, from the color heatmap, it can be seen that the accumulation of volatile aroma was most affected in CK, followed by the YJ treatment. The secondary metabolites, namely volatile aroma compounds, with the highest content in berry were C6 aldehydes and C6 alcohols, and the ester content was relatively low.
Figure 6Changes in fruit quality difference of grape fruit after GB03 microbial inoculant treatment. (**a**) TSS, (**b**) Fructose, (**c**) Glucose, (**d**) Reducing sugar, (**e**) Tartaric Acid, (**f**) Malic Acid, (**g**) Gluconic Acid, (**h**) Lactic Acid, (**i**) Titratable Acid, (**j**) pH, (**k**) Sugar-acid ratio, (**l**) Total anthocyanin, (**m**) Total phenol, (**n**) Total tannin. Different letters represent significant differences between samples under different treatments, calculated through one-way ANOVA, with *p* < 0.05 level. The bars in the graph are error lines.
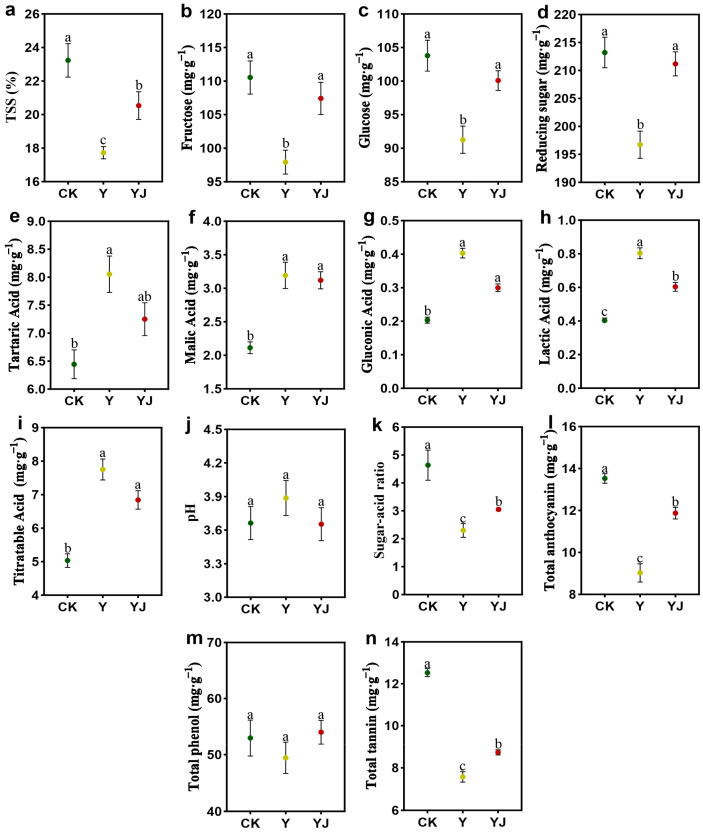



PCA analysis of the 15 key differential metabolites screened out showed that the contribution rate of the first principal component is 84.70%, and the contribution rate of the second principal component is 9.21% (Figure 8). The *X*-axis effectively distinguishes each treatment, while Y treatment was distributed on the left side of the coordinate axis, and the CK and YJ treatment were concentrated on the right side of the coordinate axis, indicating a high degree of similarity between the two treatments. At the same time, the scatter points corresponding to each treatment showed good clustering within treatment, indicating good repeatability within treatment. Among them, the CK and YJ treatment had more distribution of aromatic compounds such as hexanal, hexan-1-ol, 3-hexenal, trans-2-hexenal, 2-hexenal, (Z)-hex-3-en-1-ol, 3-Methyl-3-heptanol, benzyl alcohol, 2-ethyldecan-1-ol, and Decyl Decanoate.

### 3.9. Correlation Analysis Between Soil Physicochemical Indicators and Fruit Quality in ‘Cabernet Sauvignon’ Vineyards Under Salt Alkali Stress After Treatment with GB03 Microbial Inoculant

The correlation analysis between soil physicochemical properties and berry quality indexes shows a close relationship between soil physicochemical properties and berry quality (Figure 9). The electrical conductivity in soil was significantly positively correlated with the malic acid content in berries, and significantly negatively correlated with the content of benzoyl alcohol and Decyl Decanoate in berries. The soil pH was significantly positively correlated with the content of titratable acid and trans-2-Hexen-1-ol in berries, and significantly negatively correlated with the total tannin content in berries. The Na^+^ content in the soil was significantly negatively correlated with the single ear weight, 100-grain weight, yield, total anthocyanin, and hexanal content of the berry, extremely significantly negatively correlated with the 3-Methyl-3-heptanol content of the berry, and significantly positively correlated with the L* value of the berry. The K^+^ content in soil was significantly positively correlated with cluster weight, yield, total anthocyanin, 3-Methyl-3-heptanol, and hexanal content, and significantly negatively correlated with berry L* value; The total salt content was significantly positively correlated with the content of Titratable acid and trans-2-Hexen-1-ol in berries, and significantly negatively correlated with the content of Total tannin and 2,6-Di-tert-butyl-4-methylphenol in berries; The organic matter content and ammonium nitrogen content were significantly positively correlated with berry cluster weight, glucose content, yield, and total anthocyanin content. There was a highly significant positive correlation between nitrate nitrogen content and hexan-1-ol content in berries. There was a significant positive correlation between urease content in soil and (Z)-hex-3-en-1-ol content in berry. The nitratase content in soil was highly significantly positively correlated with the total tannin content in berry, and negatively correlated with the trans-2-Hexen-1-ol content in berry.

## 4. Discussion

The properties of soil have a significant impact on the plant growth and berry quality [8]. Soil quality and fertility are determined by the physical, chemical, and biological characteristics of the soil [52]. These include soil organic carbon, pH, macronutrients and micronutrients required by plants, microbial communities, soil enzymes, etc. This study indicates that salt alkali stress causes an increase in total salt and ion content in grape root soil, as well as an increase in soil conductivity and pH. Previous studies have shown that high salinity and pH caused by salt alkali stress have a significant inhibitory effect on soil enzyme activity [21,22,53]. Our results indicate that high salt alkali stress reduces the activities of soil sucrase, soil urease, nitrate reductase, and nitrite reductase. Soil organic matter is one of the main sources of plant nutrition, playing a crucial role in maintaining soil health and supporting robust plant growth. Our results indicate that salt alkali stress reduces soil organic matter content, which is consistent with the results reported by Morrissey et al. [54]. Soil sucrase can increase the availability of soluble nutrients in soil, and its activity is closely related to the transformation of organic matter and respiration intensity. Therefore, the decrease in sucrase activity caused by salt-alkali stress inhibits the transformation of organic matter in the soil, leading to a decrease in soil nutrients and fertility, and limiting nutrient absorption by grape plants. Nitrogen is an important element for plant photosynthesis and growth. Urease, nitrate reductase, and nitrite reductase are key enzymes involved in nitrogen cycling in soil. Saline alkali stress reduced the activity of these three enzymes and the content of ammonium and nitrate nitrogen, leading to a decrease in soil nitrogen utilization efficiency. Therefore, high salinity and pH affect the transformation of soil carbon and nitrogen, leading to a decrease in organic matter and nutrient content, a decrease in soil fertility, and ultimately a decrease in grape berry yield and quality. However, the application of GB03 bacterial fertilizer can alleviate the inhibition of soil enzyme activity by salinity and pH stress, increase organic matter and nitrogen in the soil, and thus improve soil nutrients.

Soil microorganisms are the core driving force for the formation and sustainable development of soil fertility. Soil organic matter is mainly decomposed by microorganisms, and the release of nutrients is largely controlled by microbial biomass [55,56]. It is well known that soil properties affect the soil microbial community and composition, therefore the relationship between soil properties and soil microorganisms is mutual [13]. This study showed that high pH and salinity induced by salinity stress al-tered soil characteristics, affected soil microbial habitats and increased the diversity and abundance of bacteria and fungi in the soil. Similar to our findings, Zhang et al. found that salinity is the key to the structural differentiation of microbial communities. With the in-crease of soil salinity, the relative abundance of 270 lineage types belonging to halophiles, *Alpha-proteobacteria*, *Gamma-proteobacteria* and other flora continues to increase, showing a preference for high salt ecological niches’. In addition, we have checked and modified other similar issues. [57]. The application of Bacillus subtilis microbial agents significantly increased the diversity (Simpson and Shannon) and richness (Ace, Chao) index of bacterial and fungal communities in saline soil which indicated that Bacillus subtilis not only has strong adaptability to high saline alkali environments, but also can increase the diversity of soil microbial communities in saline alkali environments. Microbial phylum level analysis indicates that salt alkali treatment increases the dominant bacterial group *Acidobacterium*. The abundance of *Acidobacterota*, *Bdellovibrionota*, and *Gemmatimonadota* indicates that the three types of bacteria have better adaptability in saline alkali environments and *Acidobacterium* and Bacillus are able to promote plant growth by producing a series of plant growth promoting substances, such as plant hormones (such as gibberellins, auxins, etc.), lysozyme, and polysaccharides [58]. These substances not only stimulate root development and nutrient absorption, but also contribute to carbon, nitrogen, and sulfur cycling in the plant soil ecosystem. Bacillus subtilis can regulate soil nutrients and alter the structure of soil microbial communities [59], which is consistent with our findings. The number of dominant bacteria and fungi in YJ treatment decreased compared with Y treatment, which may be due to the fact that GB03 reducing the stimulation of saline alkali stress on soil, thereby reducing the abundance of dominant bacterial groups such as *Acidobacteria* and *Bacillus* in rhizosphere soil.

The ion toxicity and pH stress caused by high salinity and alkalinity have an impact on the growth and development of grape plants, photosynthesis, and organic acid metabolism [60], leading to poor berry quality and reduced yield [24,61]. Li et al. [62] reported that under high salinity (150 mM), the soluble sugar content in grape berries decreased, while organic acid levels increased, anthocyanin content increased, and TSS decreased, which is consistent with our research results. Plants synthesize organic matter through photosynthesis, providing raw materials for the formation of sugars and organic acids in berries. In tomato berries, salinity regulates the activity of sucrose metabolism enzymes by controlling gene expression, thereby increasing the accumulation of fructose and glucose [63]. Therefore, the decrease in soluble sugar content, TSS, cluster weight, and 100-grain weight in the berry, as well as the increase in organic acid content, may be due to the interference of high salinity and high pH on the photosynthesis of ‘*Cabernet Sauvignon*’ grapes, altering sugar metabolism related genes and enzyme activity [60]. Research has shown that PGPR-treated plants can enhance photosynthesis and increase crop biomass. The *Bacillus velezensis* strain GB03 can enhance the photosynthetic capacity of Arabidopsis by increasing its photosynthetic efficiency and chlorophyll content [64]. Introducing GB03 into soil can lead to the accumulation of wheat biomass [65]. Therefore, the application of GB03 bacterial agent alleviated the adverse effects of salt alkali stress on grape photosynthesis and organic acid metabolism, thereby increasing grape cluster weight, 100-grain weight, soluble sugars, and reducing organic acids under alkaline stress. Anthocyanins and tannins are one of the main flavonoids, which respond to various abiotic stresses such as salt stress [66]. The *MYB*, *bHLH*, and WDR complexes play a crucial role in promoting anthocyanin production through the phenylpropanoid biosynthesis pathway [67]. In many plant species, including grapevines, *MYB*, *bHLH*, *WDR*, and other genes involved in anthocyanin production respond to abiotic stimuli [68]. Research on licorice has shown [69] that *Bacillus velezensis* increases flavonoid biosynthesis under salt stress by mediating transcription and metabolic pathways to enhance salt stress tolerance. Therefore, the application of GB03 bacterial agent may alleviate the expression of flavonoid synthesis related genes and the accumulation of metabolites in ‘*Cabernet Sauvignon*’ grapes, thereby increasing the content of anthocyanins and tannins in grape berries under salt alkali stress. The aroma of grapes is derived from aromatic volatiles and glycoside aroma compounds, including alcohols, esters, acids, terpenes, ketones and aldehydes. High concentration (150 mM) NaCl stress treatment increased the aroma components of Kyoho grape berries [54]. However, our results indicate that high salt alkali stress (300 mM) reduced the aroma components of ‘*Cabernet Sauvignon*’ grape berries. This may be due to the high salt concentration and high pH stress caused by NaHCO_3_, leading to a decrease in grape berry aroma. However, the application of GB03 bacterial agent alleviated the negative effects of salt alkali stress on grape plant growth and berry quality, which may have promoted the accumulation of aroma components in berries.

The Gansu Hexi Corridor region is located in northwestern China, at latitudes of 36–40° N and altitudes of 1500–2000 m. It has the most suitable combination of water, soil, light, and heat resources for the production of wine grapes, as well as a profound cultural heritage and long history of winemaking [70]. This region is one of the high-quality wine producing areas in China. However, more than 90% of the region’s soils are saline and desert soils, and the pH of the soils is generally higher than 8.0, with extremely high saline content, which seriously affects the normal growth of main plant varieties such as *Cabernet Sauvignon* [71]. Our results demonstrate the efficacy of the GB03 microbial inoculants in alleviating saline stress and provide guidance for viticulture in saline soils. However, this study was conducted only for one year of sampling at the same site to obtain experimental data, whereas soil physicochemical properties, soil microbial diversity, and grapevine berry yield and quality may be affected by different climates and rainfall each year. Therefore, in the future, we will conduct multi-year, larger-scale continuous sampling to obtain more valid data and results to provide a stronger scientific basis for microbial agents to improve crop yield and quality in saline soils.

## 5. Conclusions

The application of GB03 microbial agent under salt alkali stress could reduce the electrical conductivity, soil pH, Na^+^, and total salt content of vineyard soil, increase the K^+^, organic matter, ammonium nitrogen, and nitrate nitrogen content of soil, enhance the activities of invertase, urease, and nitratase in soil. The application of GB03 microbial agent under salt alkali stress could enhance the Ace, Chao, Simpson, and Shannon indices of true fungi in soil, increase the number of dominant bacteria and fungi, and reduce the number of rare bacteria and fungi, increase the number of dominant bacterial strains, such as *Acidobacteriota*, *Bdellovibrionota*, and *Gemmatimonadota*, recruit fungal communities such as *Linatneria* and *Diehliomyces*, increase the relative abundance of bacterial genera such as *Bacillus*, *Pelagibius*, *Filobassidium*, and *Mortierella*. Most importantly, the application of GB03 microbial agent under salt alkali stress improve the quality and yield of grape berries under salt stress which plays an important role in improving saline alkali soil (Figure 10).

## Figures and Tables

**Figure 1 foods-14-00711-f001:**
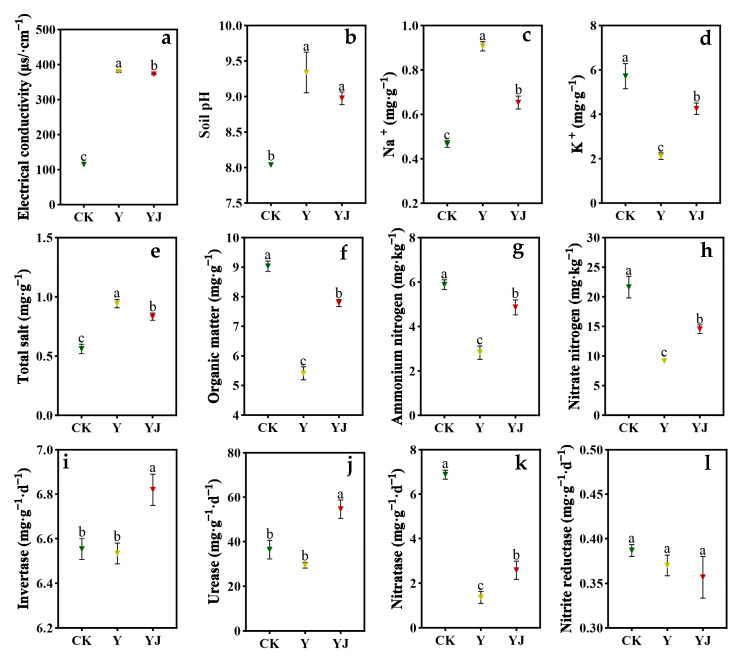
Physicochemical properties and soil enzyme activities of vineyard soils under different treatments. (**a**) Electrical conductivity, (**b**) Soil pH, (**c**) Na^+^, (**d**) K^+^, (**e**) Total salt, (**f**) Organic matter, (**g**) Ammonium nitrogen, (**h**) Nitrate nitrogen, (**i**) Invertase, (**j**) Urease, (**k**) Nitratase, (**l**) Nitrite reductase. Different letters represent significant differences between samples under different treatments, calculated through one-way ANOVA, with *p* < 0.05 level. The bars in the graph are error lines.

**Figure 2 foods-14-00711-f002:**
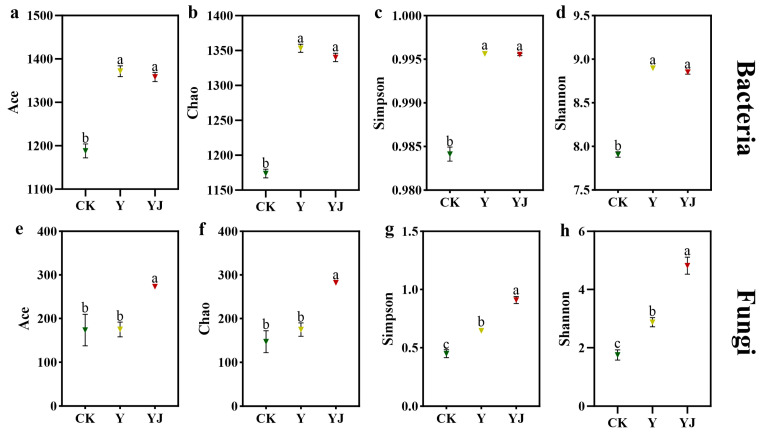
Alpha diversity of bacteria and fungi in grape rhizosphere soil under different treatments. Bacteria community richness: (**a**) Ace, (**b**) Chao. Bacteria community diversity: (**c**) Simpson, (**d**) Shannon. Fungi community richness: (**e**) Ace, (**f**) Chao. Fungi community diversity: (**g**) Simpson, (**h**) Shannon. Different letters represent significant differences between samples under different treatments, calculated through one-way ANOVA, with *p* < 0.05 level. The bars in the graph are error lines.

**Figure 3 foods-14-00711-f003:**
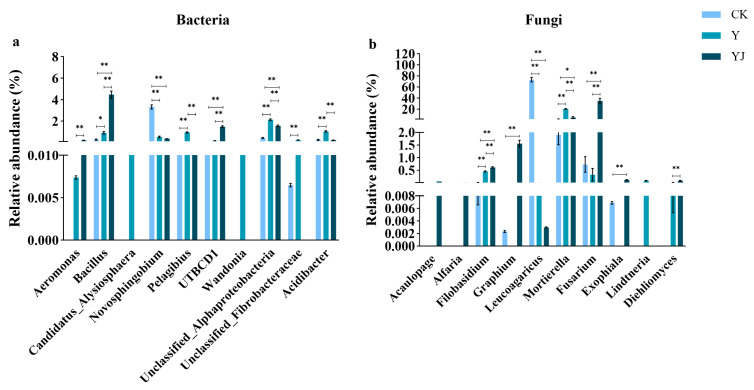
Effects of different treatments on the relative abundance of microbiota at the genus Level in the rhizosphere soil of grapevines. (**a**) Bacteria, (**b**) Fungi. The asterisk above the bar represents a significant difference at the level of *p* < 0.05, and the two asterisks represent a significant difference at the level of *p* < 0.01.

**Figure 7 foods-14-00711-f007:**
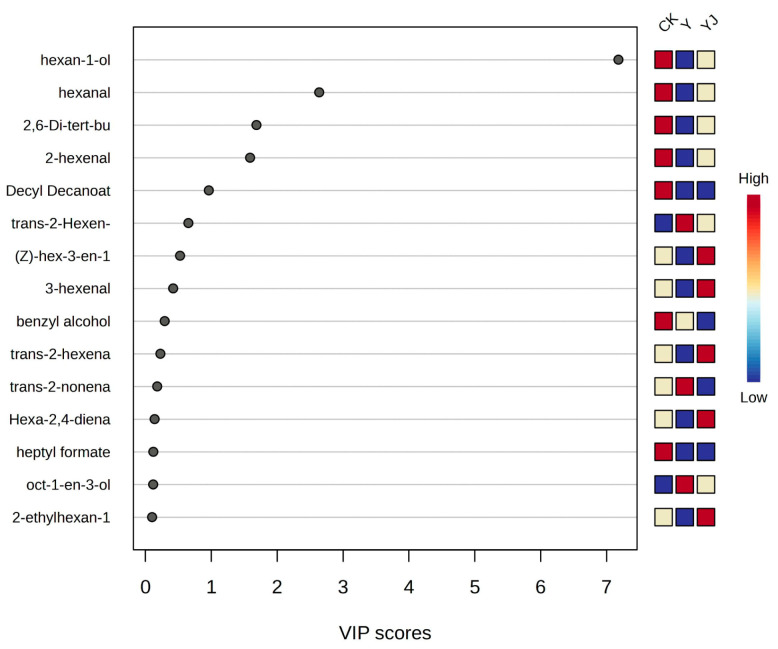
Partial least squares discriminant analysis of volatile aroma compounds in grape berries under different treatments.

**Figure 8 foods-14-00711-f008:**
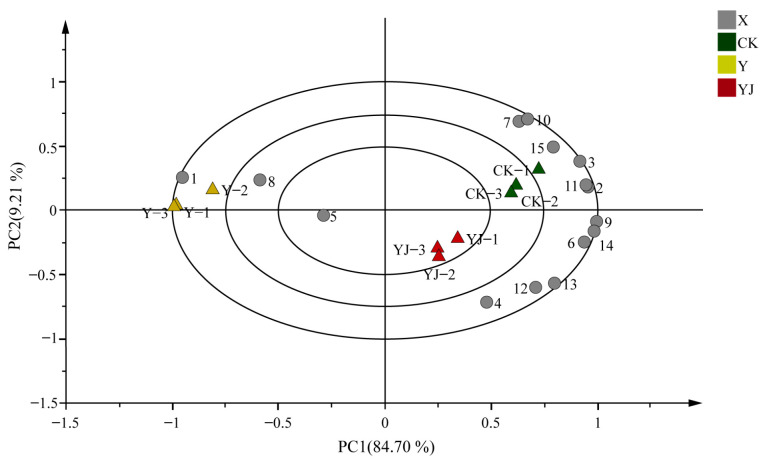
Principal component analysis of key volatile aroma compounds in grape berries under different treatments 1–15 respectively represent: 1, pentan-1-ol, 2, 3-Methyl-3-heptanol, 3, hexan-1-ol, 4, (Z)-hex-3-en-1-ol, 5, trans-2-Hexen-1-ol, 6, 2-butyl-1-octanol, 7, benzyl alcohol, 8, 2-Nonen-1-ol, 9, 2-ethyldecan-1-ol, 10, Decyl Decanoate, 11, hexenal, 12, 3-hexenal, 13, trans-2-hexenal, 14, 2-hexenal, 15, 2,6-Di-tert-butyl-4-methylphenol.

**Figure 9 foods-14-00711-f009:**
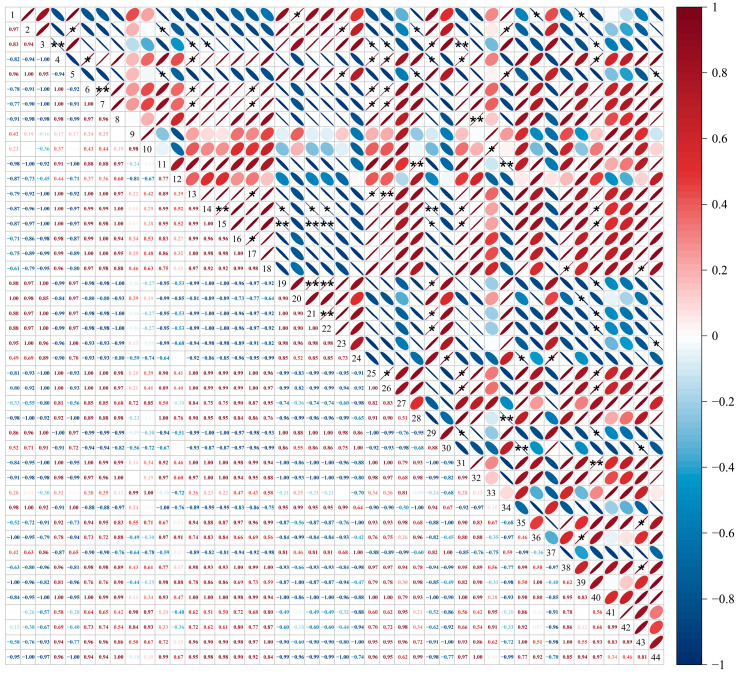
Correlation analysis between soil microorganisms and physicochemical parameters in vineyards and grape quality and volatile matter indicators. 1–44 respectively represent: 1, Electrical conductivity, 2, Soil pH, 3, Soil Na^+^, 4, Soil K^+^, 5, Total salt, 6, Organic matter, 7, Ammonium nitrogen, 8, Nitrate nitrogen, 9, Invertase, 10, Urease, 11, Nitratase, 12, Nitrite reductase, 13, Cluster weight, 14, 100-grain weight, 15, TSS, 16, Fructose, 17, Glucose, 18, Reducing sugar, 19, Tartaric Acid, 20, Malic Acid, 21, Gluconic Acid, 22, Lactic Acid, 23, Titratable Acid, 24, berry pH, 25, Yield, 26, Total anthocyanin, 27, Total phenol, 28, Total tannin, 29, L*, 30, pentan-1-ol, 31, 3-Methyl-3-heptanol, 32, hexan-1-ol, 33, (Z)-hex-3-en-1-ol, 34, trans-2-Hexen-1-ol, 35, 2-butyl-1-octanol, 36, benzyl alcohol, 37, 2-Nonen-1-ol, 38, 2-ethyldecan-1-ol, 39, Decyl Decanoate, 40, hexanal, 41, 3-hexenal, 42, trans-2-hexenal, 43, 2-hexenal, 44, 2,6-Di-tert-butyl-4-methylphenol. Positive correlations are shown in red and negative correlations in blue, with smaller ellipses indicating higher correlations. The numbers in the graphs indicate the correlation coefficients, with positive values indicating positive correlations and negative values indicating negative correlations. * indicates significant correlation at the 0.05 level, ** indicates significant correlation at the 0.01 level.

**Figure 10 foods-14-00711-f010:**
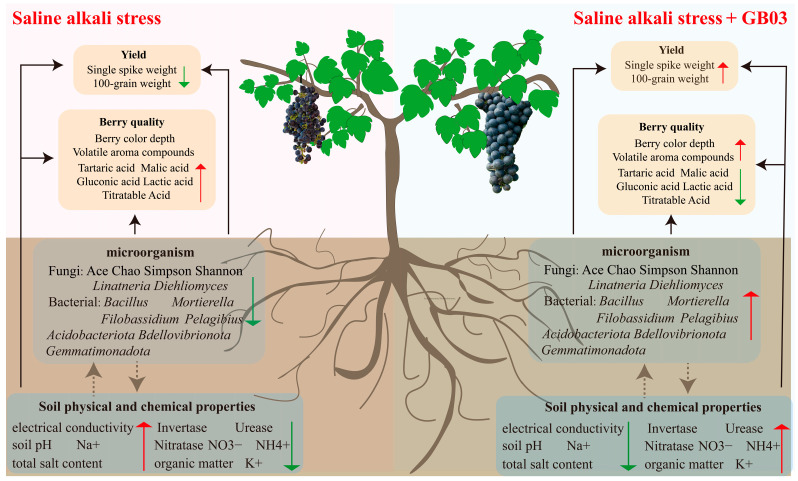
Red arrows indicate an increase in each indicator, while green arrows indicate a decrease. The gray dashed arrows represent a potential relationship that has not been further validated, whereas the black solid arrows represent a confirmed relationship.

**Table 1 foods-14-00711-t001:** Grape berry color indicators under different treatments.

Treatments	L*	a*	b*	C*
CK	32.07 ± 0.53 c	2.35 ± 0.17 a	3.15 ± 0.13 a	15.47 ± 0.06 a
Y	37.79 ± 1.62 a	2.08 ± 0.14 a	3.01 ± 0.11 a	13.44 ± 1.08 a
YJ	34.78 ± 0.92 b	2.13 ± 0.14 a	2.92 ± 0.22 a	13.17 ± 1.87 a

Note: Lowercase letters represent significant differences among sampling treatments (*p* < 0.05).

## Data Availability

The original contributions presented in the study are included in the article/Appendix A, further inquiries can be directed to the corresponding authors.

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
