# Peer review of "Microbial Inoculant GB03 Increased the Yield and Quality of Grape Fruit Under Salt-Alkali Stress by Changing Rhizosphere Microbial Communities"

_foods, 2025, doi:10.3390/foods14050711_

Round 1

Reviewer 1 Report

Comments and Suggestions for Authors

The paper discussed Microbial inoculant GB03 increased the yield and quality of grape fruit under salt-alkali stress by changing rhizosphere microbial communities. The application of GB03 microbial agent under salt alkali stress could reduce the electrical conductivity, soil pH, Na+ , and total salt content of vineyard soil, increase the K+ ,  organic matter, ammonium nitrogen, and nitrate nitrogen content of soil, enhance the activities of invertase, urease, and nitratase in soil.  Some comments are as follows:

1.What is the limitation of applying GB03 microbial agent?

2.Line 76: The authors wrote "The application of Plant Growth Promoting Rhizobacteria (PGPR) has been recognized as an effective measure to increase agricultural productivity and improve soil and environmental health [25]. Please explain the mechanism of how this method increase agricultural productivity and improve soil and environmental health.

3.Line 95: The authors wrote " GB03 could promote plant growth and increase crop   yield".  Please explain the mechanism of how GB03 promote plant growth and increase crop   yield as it changes the physicochemical properties of soil, enzyme activity, microbial community diversity under salt-alkali stress, and improve fruit quality and aroma components.  

4.Line 118: There is a large temperature difference between day and night. Thus, Thermal unit requirement of grape and heat use efficiency   values must be provided under these conditions.

5. The authors wrote " GB03 microbial agent is developed by Lanzhou University and Gansu Agricultural 121 University, and produced by Gansu HongYuan Biotechnology Co., Ltd (The main com- 122 ponents are Bacillus velezensis, effective viable count ≥ 108 CFU/g). Please give details  about characteristics of GB03 microbial agent.

6.Line 126: The authors wrote " Three rows of grapes with similar growth were selected".  What is the method to identify the grapes with similar growth?

 7. Please provide the specifications of the pH   meter.

8. Line  243: Please give details  about  NIST.

9. Line  259: put respectively after 11.70%.

10. Line  282:  How do you measure Alpha diversity  and the diversity index.

11. Line  292:  what is the Binary jaccard distance.

12.  Line  298:  How do you  determine  cumulative contribution rates.

13. In materials and method section, there were no method to determine, fructose, glucose, ,  sugar-acid ratio, and the content of total  tannin.

14.In Table 1, what is the values of Grape berry color indicators under control treatment.

15.  Line  425:   what is VIP score .

16.  Line  571:   please explain The MYB, bHLH, and WDR.

17.  Line  599:   please explain Ace, Chao, Simpson, and Shannon indices.

18. Line 604: The authors wrote " Most importantly, the application of  GB03 microbial agent under salt alkali stress improve the quality and yield of grape berries under salt stress which plays an important role in improving saline alkali soil". In the text there were no data for the control treatment to show the color differences among treatments and also yield.

19. What is the novelty of the study.

Comments on the Quality of English Language

Some improvements are required for English and figures.

Reviewer 2 Report

Comments and Suggestions for Authors

Reviewer comments for the manuscript titled ''Microbial inoculant GB03 increased the yield and quality of grape fruit under salt-alkali stress by changing rhizosphere microbial communities'' (ID: foods-3441049)

 The authors mainly discuss the impact of a microbial inoculant on several traits of grapevine and various physicochemical properties and microbial communities of soil under saline stress conditions. The topic is interesting, accompanied with relevant results which contribute to the literature. However, there are some issues that need to be addressed for a further improvement of the quality of the manuscript. 

 General concept comments

To what extent do authors consider that their methodological procedure represents reality? Are there limitations? The authors collected their data from a particular site in the Hexi Corridor, China for one growing season, if I understand well. A larger set of data, spatiotemporally, could be useful for testing the usefulness of the applied methodology, revealing possibly more information relevant to the examined topic. These issues must be discussed.

The 'Introduction' section is considered satisfactory, but the inclusion of a hypothesis would be an added value.

The section 'Materials and Methods' should be enriched in order to be more comprehensive. Specifically, the authors should provide information about the calculation of the soil Ace, Chao, Simpson, and Shannon indices of soil microorganisms since these indices are crucial components of their study.

  Specific comments

1.        Lines 45-48. The quality of the berry... ity[6,7]. Rephrase.

2.        Lines 53-54. are microscopic in the rhizosphere microenvironment Is this necessary?

3.    Line 82. agent to wheat Wheat is a generic name. Provide the complete scientific name. Check for similar cases throughout the text.

4.       Lines 112-119. The soil is neutral to... between day and night. Provide at least one reference source.

5.       Line 123. Bacillus velezensis Italicize. Check for similar cases.

6.    Line 130-131. NaCl + NaHCO3 mixed salt solution What are the percentages of 'NaCl' and 'NaHCO3' in this 'mixed salt solution'?     

7.        Line 142. when they were ripe When (month and year)?

8.        Lines 149-150. Collect soil samples... by Zhang et al.[38]. Rephrase.

9.        Lines 156-157. a conductivity meter...pH meter Provide the name of the model, and the name, city and country of manufacturer for both instuments.

10.    Line 164. by kit  Provide the name of the 'kit'.

11.    Line 175. PCR What is 'PCR'? First define and then abbreviate. Check for similar cases.  

12.    Fig. 1. Mark the twelve images of 'Fig. 1' with 'a', 'b' etc. and rewrite correctly in its legend. Also figures (and tables) should be self-explanatory. For example, what do the bars in the main body of 'Fig. 1' represent? Check for similar cases.

13.    Line 293. (Fig. S1). Where is 'Fig. S1'? Check for similar cases.

14.  Lines 362-365. Rephrase the legend of 'Fig. 3' including 'A' (first image) and 'B' (second image).

15.    Lines 529-530. ...ecological niches[53]. This is similar to our findings, where high pH... Taking this text into account, first the authors mention the findings of other researchers and then their own findings. The authors should mention first their own findings and then the findings of others. Check for similar cases throughout the 'Discussion'. 

16.    Line 591. The black dashed arrows 'black' or grey?

Round 2

Reviewer 1 Report

Comments and Suggestions for Authors

No comments

Author Response

Dear Prof: 
Thank you for your letter and for the reviewers’ comments concerning our manuscript entitled “Microbial inoculant GB03 increased the yield and quality of grape fruit under salt-alkali stress by changing rhizosphere microbial communities (id:foods-3441049)”. Those comments are all valuable and very helpful for revising and improving our paper, as well as the important guiding significance to our researches. 

Reviewer 2 Report

Comments and Suggestions for Authors

Reviewer comments for the manuscript titled ''Microbial inoculant GB03 increased the yield and quality of grape fruit under salt-alkali stress by changing rhizosphere microbial communities'' (ID: foods-3441049)

 The manuscript is a new revised version and mainly discuss the impact of a microbial inoculant on several traits of grapevine and various physicochemical properties and microbial communities of soil under saline stress conditions. The authors wrote 'The main corrections in the paper and the responds to the reviewer’s comments are as flowing:' but they addressed (quite satisfactorily) only  the specific comments no. 1-16 (from my previous review) in their cover letter and revised version. The general concept comments from my previous review were not addressed at all. Therefore, there are still issues that need to be addressed for the improvement of the quality of the manuscript. In my new review, I keep my recommendation 'Reconsider after major revisions...' until the clarification of the remaining issues. The aforementioned clarification should include the point by point authors' response to my current comments with a brief explanation and line numbers in the new revised manuscript.

The authors must consider the following:

 1. To what extent do authors consider that their methodological procedure represents reality? Are there limitations? The authors collected their data from a particular site in the Hexi Corridor, China for one growing season, if I understand well. A larger set of data, spatiotemporally, could be useful for testing the usefulness of the applied methodology, revealing possibly more information relevant to the examined topic. These issues must be discussed.

2. The 'Introduction' section is considered satisfactory, but the inclusion of a hypothesis would be an added value.

3. The section 'Materials and Methods' should be enriched in order to be more comprehensive. Specifically, the authors should provide information about the calculation of the soil Ace, Chao, Simpson, and Shannon indices of soil microorganisms since these indices are crucial components of their study.

4. Fig. 2. Mark the eight images of 'Fig. 2' with 'a', 'b' etc. and rewrite correctly in its legend.

5. Fig. 4, Fig. 9. Clarify these figures in order to be more understandable.
